# Integrating human endogenous retroviruses into transcriptome-wide association studies highlights novel risk factors for major psychiatric conditions

Rodrigo R. R. Duarte [1,2] ✉, Oliver Pain [3], Matthew L. Bendall [2], Miguel de Mulder Rougvie[2], Jez L. Marston[2], Sashika Selvackadunco[3,4], Claire Troakes [3,4], Szi Kay Leung [5], Rosemary A. Bamford[5], Jonathan Mill [5], Paul F. O'Reilly [6], Deepak P. Srivastava [3,7], Douglas F. Nixon [2,8,9] & Timothy R. Powell [1,2,9] ✉

Human endogenous retroviruses (HERVs) are repetitive elements previously implicated in major psychiatric conditions, but their role in aetiology remains unclear. Here, we perform specialised transcriptome-wide association studies that consider HERV expression quantified to precise genomic locations, using RNA sequencing and genetic data from 792 *post-mortem* brain samples. In Europeans, we identify 1238 HERVs with expression regulated in *cis*, of which 26 represent expression signals associated with psychiatric disorders, with ten being conditionally independent from neighbouring expression signals. Of these, five are additionally significant in fine-mapping analyses and thus are considered high confidence risk HERVs. These include two HERV expression signatures specific to schizophrenia risk, one shared between schizophrenia and bipolar disorder, and one specific to major depressive disorder. No robust signatures are identified for autism spectrum conditions or attention deficit hyperactivity disorder in Europeans, or for any psychiatric trait in other ancestries, although this is likely a result of relatively limited statistical power. Ultimately, our study highlights extensive HERV expression and regulation in the adult cortex, including in association with psychiatric disorder risk, therefore providing a rationale for exploring neurological HERV expression in complex neuropsychiatric traits.

Psychiatric disorders such as schizophrenia, bipolar disorder, major depressive disorder, attention deficit hyperactivity disorder, and autism spectrum conditions have a substantial genetic component[1]. Genome-wide association studies (GWAS) have highlighted a polygenic architecture underlying susceptibility to these conditions, meaning that many loci across the genome incrementally contribute to risk. As associated variants are mostly non-coding and therefore assumed to impact the regulation of local genes, transcriptome-wide association studies (TWAS) were developed to aid the identification of gene expression signatures associated with susceptibility[2]. They represent a powerful approach that has the potential to reveal insights into disorder aetiology and lead to the identification of new drug targets[3]. TWAS draw power from large genetic association studies to test risk variants for association with the expression of local genes in

---

relevant tissues, after accounting for genetic structure (linkage disequilibrium). While this method has facilitated the identification of genes and biological processes associated with major psychiatric conditions[4–6], it has also largely overlooked the expression of repetitive elements like human endogenous retroviruses (HERVs), in relation to susceptibility.

HERVs are "non-coding" sequences comprising of genetic material that originated from the infection of germ cells with ancient retroviruses during evolution, which now constitute approximately 8% of the human genome[7–9]. After the initial infections took place, these sequences inserted in the genome and multiplied themselves using a 'copy-and-paste' mechanism known as retrotransposition. At present, there is no evidence that these elements are currently retrotransposing, and studies suggest the majority of HERV insertions occurred over ~1.2 million years ago[10,11]. Instead, they have been hypothesised to regulate neighbouring genes, as most HERV sequences comprise of solitary viral promoters known as *long terminal repeats* (LTRs)[9,12]. However, many sequences additionally contain remnants of viral genes (e.g., *gag, pol, env*) that may encode additional biological functions, other than just regulating gene expression locally. For example, HERVs from the families W and FRD encoding *env* play a fundamental role in cellular fusion during the formation of the placenta and are now annotated as the syncytin-1 and syncytin-2 genes, respectively[13]. Critically, 14,968 HERV transcriptional units comprising of ancient viral genes flanked by LTRs have been annotated in the reference genome, from across 60 HERV families[14]. Although HERVs have been implicated in major psychiatric conditions[15–20], most studies precede the comprehensive genomic annotation of these sequences. These studies also relied on methods that aggregate family-level expression data, such as Western blotting, reverse transcriptase quantitative PCR (RT-qPCR), or microarrays, and most also analysed very small sample sizes, meaning they were underpowered for the investigation of complex polygenic traits[1]. Finally, by employing case-control study designs, they were more likely to capture expression changes elicited by environmental factors associated with a psychiatric diagnosis, such as smoking or treatment[21].

Here, we use a TWAS approach that considers neurological HERV expression estimated to precise genomic locations, to identify expression signatures associated with psychiatric conditions, while circumventing the limitations more prevalent in traditional case-control studies. Due to the inclusion of global HERV expression, or the 'retrotranscriptome', in this analysis, we call this approach a 'retrotranscriptome-wide association study' (rTWAS). We identify extensive HERV expression and regulation in the adult cortex, including in association with genetic risk for psychiatric disorders. We also detect co-expression networks linking the expression of canonical genes with HERVs, allowing us to broadly infer the function some specific HERVs may play in neurobiology. This work provides a rationale for exploring neurological HERV expression in complex neuropsychiatric traits.

## Results
### *Cis*-heritable expression in the dorsolateral prefrontal cortex
A summary of our approach is outlined in Fig. 1. The number of HERVs and canonical genes detected as consistently expressed in the DLPFC samples from donors of European ($N = 563$) and African ($N = 229$) ancestries is provided in Table 1. Table 1 also shows the number of genetic features detected as expressed in the autosomes, as only these can be cross-referenced with publicly available GWAS results in a standard TWAS approach. The table also includes the number of genetic features showing significant *cis*-heritable expression according to a likelihood ratio test (nominal $P < 0.01$). Interestingly, of the 4645 HERVs expressed in the African sample, 4463 (96%) were also detected in the European cohort. However, of the 852 HERVs exhibiting *cis*-heritable expression in Africans, only 534 (63%) displayed

*cis*-heritable expression in Europeans. Although caution is needed when interpreting these results due to variations in statistical power between the cohorts, these figures preliminarily suggest ancestry-specific differences in HERV expression regulation.

### Retrotranscriptome-wide association studies
We initially investigated psychiatric traits explored in European cohorts, as these represent the most well-powered genetic studies published to date, using the SNP weights calculated with the European subset of the CommonMind Consortium. In total, we identified 26 HERV expression signatures associated with psychiatric disorder susceptibility. More specifically, for schizophrenia, the rTWAS identified 163 Bonferroni-significant risk expression signatures, of which 15 (9%) pertained to HERVs, including 9 positively regulated and 6 negatively regulated features, in association with genetic risk (Fig. 2A). The top HERV expression association signals originated from the major histocompatibility complex (MHC) locus, on chromosome 6p21-22 (ERV316A3_6p22.1b, $Z = -8.75$, $P = 2.05 \times 10^{-18}$), and chromosome 2q33 (ERV316A3_2q33.1 g, $Z = -7.13$, $P = 1.03 \times 10^{-12}$). This analysis replicated schizophrenia expression signatures identified previously in a TWAS that considered *cis*-heritable expression in a subset of the CMC cohort[2] (e.g., *NAGA*, $Z = 7.74$, $P = 9.58 \times 10^{-15}$; *SNAP91*, $Z = 4.80$, $P = 1.61 \times 10^{-6}$; *TAOK2*, $Z = -7.44$, $P = 1.04 \times 10^{-13}$) and in the developing brain[22] (e.g., *SF3B1*, $Z = 6.99$, $P = 2.78 \times 10^{-12}$; *MAPK3*, $Z = 5.68$, $P = 1.39 \times 10^{-8}$; *FURIN*, $Z = -8.44$, $P = 3.11 \times 10^{-17}$).

For other traits, we identified fewer expression signatures associated with risk, likely because of the smaller cohorts analysed in the GWAS, or the reduced heritability of the traits. For instance, for bipolar disorder, we identified 47 expression signatures associated with susceptibility, of which only two (4%) were HERVs (MER4_20q13.13, $Z = 5.04$, $P = 4.73 \times 10^{-7}$; PRIMA41_9q34.3, $Z = 4.61$, $P = 4.07 \times 10^{-6}$; Fig. 2B). Interestingly, MER4_20q13.13 was also a HERV identified in the schizophrenia rTWAS, with the same direction of effect ($Z = 9.95$, $P = 8.15 \times 10^{-5}$). For major depressive disorder, we identified 29 signatures, of which 9 (31%) were HERVs, including five on chromosome 1p31, two on chromosome 9p23, and one each on chromosomes 3p21 and 14q24 (Fig. 2C). For attention deficit hyperactivity disorder and autism spectrum conditions, we identified seven and one expression signatures associated with risk, respectively, although none corresponded to HERVs. All significant expression signatures (Bonferroni $P < 0.05$), including those pertaining to canonical genes, are shown in Supplementary Data 1.

### Conditional analyses
We performed conditional analyses within FUSION to identify jointly and conditionally independent associations, allowing us to isolate HERV expression associations that were independent from the expression of surrounding canonical genes and that further explained the GWAS signal in their loci. For schizophrenia, we identified 91 conditionally independent associations, of which 6 (7%) corresponded to HERVs. These included MER4_20q13.13 (TWAS $P = 9.90 \times 10^{-9}$; joint $P = 1.00 \times 10^{-8}$), ERV316A3_2q33.1 g (TWAS $P = 1.00 \times 10^{-12}$; joint $P = 2.90 \times 10^{-8}$), and ERV316A3_5q14.3j (TWAS P and joint $P = 5.50 \times 10^{-6}$; Fig. 3A). For bipolar disorder, we found 30 conditionally independent associations, of which two (7%) related to HERVs, including MER4_20q13.13 (TWAS P and joint $P = 4.70 \times 10^{-7}$; Fig. 3B). For major depressive disorder, we identified 12 conditionally independent associations, of which 2 (17%) related to HERVs, including ERVLE_1p31.1c (TWAS P and joint $P = 2.90 \times 10^{-18}$; Fig. 3C). For attention deficit hyperactivity disorder and autism spectrum conditions, we identified four and one conditionally independent associations, respectively, although these pertained to canonical genes only. All joint significant expression signals, including those pertaining to canonical genes, are shown in Supplementary Data 2.

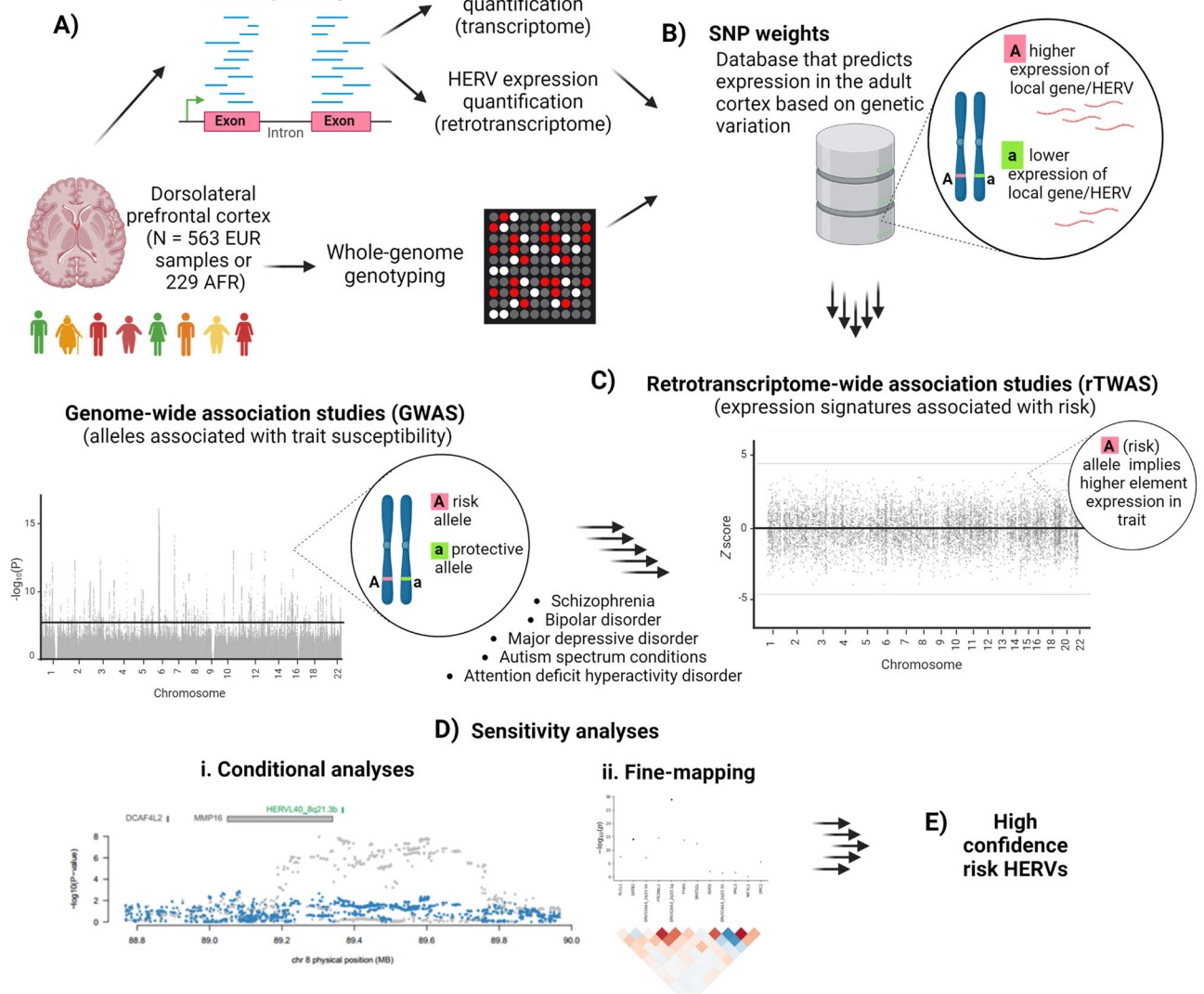

**Fig. 1 | A summary of the retrotranscriptome-wide association study (rTWAS) approach. A** RNA-sequencing and genotype data from individuals of European (EUR, *N* = 563) or African ancestry (AFR, *N* = 229) are used to construct (**B**) single nucleotide polymorphism (SNP) weights. The example depicts a genetic feature more expressed in association with the A-allele from a hypothetical local variant, relative to the alternative a-allele. **C** GWAS results are then cross-referenced with the SNP weights using a transcriptome-wide association study (TWAS) approach, to identify expression signatures associated with risk. The example illustrates that the A-allele of the hypothetical variant, associated with increased expression of the hypothetical genetic feature, is also associated with trait susceptibility. **D** Sensitivity analyses including **i.** conditional analyses and **ii.** fine-mapping then allow inference of which expression signals are considered **E** high confidence risk features, as indicated by their ability to independently explain the genetic signal at their respective loci. Created with Biorender.com. This image is published under a CC BY-NC-ND license.

## rTWAS fine-mapping

For schizophrenia, the fine-mapping analysis showed 11 HERV expression signatures that were more likely to explain the association signal at their loci relative to neighbouring genetic features (posterior inclusion probability (PIP) > 0.5). Of these, three were associated with schizophrenia in the conditional analyses and thus are considered high confidence risk HERVs (Fig. 4A). These included ERV316A3_2q33.1 g (PIP = 1.00), ERV316A3_5q14.3j (PIP = 0.98), and MER4_20q13.13 (PIP = 1.00). For bipolar disorder, we identified two HERVs with PIP > 0.50, of which one was considered independent according to the conditional analysis (MER4_20q13.13, PIP = 0.99; Fig. 4B). For major depressive disorder, we identified four HERVs with PIP > 0.50, of which one was considered independent according to the conditional analysis, despite the complex linkage disequilibrium structure in the region (ERV-LE_1p31.1c, PIP = 0.68; Fig. 4C). For attention deficit hyperactivity disorder, we identified two HERVs on chromosome 3p24 with PIP > 0.5,

namely HARLEQUIN_3p24.3 (PIP = 0.79) and HML3_3p24.3 (PIP = 0.97), but these were not significant in the conditional/joint analyses. There were no HERV expression signals with PIP > 0.50 for autism spectrum conditions. All expression signatures with PIP > 0.50, including those pertaining to canonical genes, are shown in Supplementary Data 3. A summary of the association statistics for all high confidence risk HERVs, defined as those with PIP > 0.5 in the rTWAS fine-mapping and whose expression were additionally considered independently associated with a psychiatric trait according to conditional/joint analyses, is presented in Table 2.

## Sensitivity analyses

We included individuals with a psychiatric diagnosis at the time of death in the construction of the SNP weights for the rTWASs, as the added sample size increases power to detect *cis*-regulatory effects associated with trait susceptibility. Previously, ref. 2. found consistent

**Table 1 | HERVs and canonical genes expressed in our samples**

| Ancestry | N | HERVs | | | Canonical genes | | |
|---|---|---|---|---|---|---|---|
| | | Expressed | Autosomal | Regulated in *cis* | Expressed | Autosomal | Regulated in *cis* |
| European | 563 | 4594 | 4289 (93%) | 1238 (27%) | 15,017 | 14,459 (96%) | 6956 (46%) |
| African | 229 | 4645 | 4343 (93%) | 852 (18%) | 15,015 | 14,546 (97%) | 5464 (36%) |

Percentages are relative to the total number of expressed HERVs or canonical genes, per cohort.

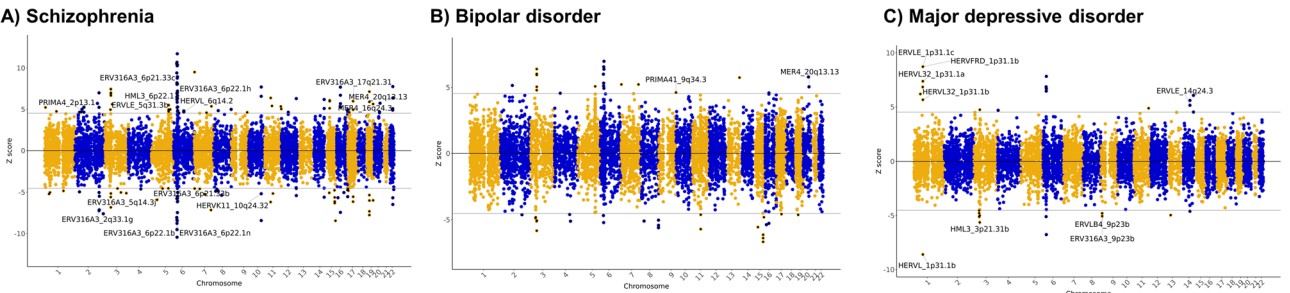

**Fig. 2 | Retrotranscriptome-wide association studies of major psychiatric disorders.** The Manhattan biplots show the expression signatures significantly associated with (**A**) schizophrenia, (**B**) bipolar disorder, and (**C**) major depressive disorder. We found no HERV expression signatures associated with attention deficit hyperactivity disorder and autism spectrum conditions, so these are omitted. The X-axis indicates genomic location, whereas the Y-axis shows Z score from the TWAS. The horizontal grey lines indicate transcriptome-wide significance, i.e., a threshold adjusted for the number of expressed features using the Bonferroni method (two-sided $P$ value cut-off = $6.10 \times 10^{-6}$). Only Bonferroni-significant HERV features are labelled.

*cis*-regulatory mechanisms governing gene expression across cases and controls in this dataset. However, to ensure that the same applies to HERV expression, we also constructed TWAS weights using expression data from unaffected controls only ($N = 242$ unaffected individuals), and compared their performance against weights constructed using the full European sample ($N = 563$). We found evidence that by adding cases alongside controls (and thus increasing sample size by 133%), we increased the detection of HERVs with a *cis*-heritable expression by 85%. We performed a schizophrenia rTWAS using the newly calculated weights and explored how the resulting Z-scores correlated with those obtained in the schizophrenia rTWAS performed with weights calculated using the full cohort. This analysis showed an extremely high correlation (Pearson's $r(5570) = 0.95$, $P < 2.2 \times 10^{-16}$), indicating that results were very similar. However, the schizophrenia rTWAS performed using weights from unaffected individuals identified 137 expression associations (Bonferroni $P < 0.05$), corresponding to a reduction of 16% in significant expression signatures, relative to those detected in the full cohort. Amongst these, ten corresponded to HERVs, of which nine were Bonferroni significant features in the analysis comprising the full cohort (Bonferroni $P < 0.05$), whereas the tenth feature was only nominally significant (HERVL18_6p22.1c, TWAS $P = 0.002$). Overall, these findings suggest that incorporating psychiatric cases in the construction of SNP weights can bolster power to detect *cis*-heritable expression features, as well as expression signatures associated with genetic risk.

**Risk HERV signatures in non-European ancestries**
To test whether the high confidence risk HERV signatures identified in Europeans may also be relevant to other populations, we performed analyses with different ancestries. First, using weights calculated in the European subset of the CMC, we analysed GWAS summary statistics obtained from analysis of diverse ancestries. These included schizophrenia GWAS summary statistics from African American, Latino, and Asian cohorts[23], and major depressive disorder GWAS results from an East Asian cohort[24]. While analysing GWAS results with SNP weights calculated in a sample with mismatched ancestry is not ideal due to differences in linkage disequilibrium structure across populations, it can still lead to informative results (e.g., ref. 25,). We observed the association

between MER4_20q13.13 and schizophrenia in the Asian cohort with nominal confidence ($Z = 2.14$, $P = 0.03$), although this would not survive multiple testing correction for the number of expression signatures tested in that rTWAS (Bonferroni $P > 0.05$). No other high confidence expression signatures were observed. We did not test additional ancestries, as we were unable to identify publicly available summary statistics from well-powered GWASs in non-European cohorts in the NIH GWAS Catalog or the Psychiatric Genomics Consortium website.

Second, we created SNP weights based on the African American subset of the CommonMind Consortium ($N = 229$) and analysed the schizophrenia GWAS results from an African American cohort[23]. We were unable to identify Bonferroni-significant expression signals, likely because this is a severely underpowered GWAS ($N = 6152$ cases, 3918 controls). For reference, however, the top rTWAS signal pertained to $QSOX1$ ($Z = -3.89$, $P = 9.92 \times 10^{-5}$), and the top HERV expression signal pertained to HERVS71_7p14.3 ($Z = -3.56$, $P = 3.67 \times 10^{-4}$).

**Characterisation of high confidence risk HERVs**
**Genomic context.** Visualisation of the log-transformed counts per million (logCPM) of the high confidence risk HERVs in the European samples shows that their expression is lower when compared to their nearest canonical genes (Fig. 5A). This is not surprising, as HERV expression in adult brain tissue is believed to be suppressed by epigenetic markers and HERV regulators such as TRIM28[26]. However, to ensure that HERV expression signals are not biased by signals originating from pre-mRNAs from local canonical genes, we assessed the extent to which HERV signals might represent specific isoforms of these canonical genes. Using HOMER[27], we found that expressed HERVs mostly belonged to intergenic and intronic regions of the genome (98%; Fig. 5B), as expected. Then we retrieved expression and strand information for canonical genes containing intronic HERVs, which showed that these genes are mostly either not expressed or located in the opposite strand of the HERV. This suggests that the majority of expressed HERVs are likely to reflect novel non-coding RNAs, rather than specific isoforms of canonical genes.

Analysis of the high confidence risk HERVs within the Integrated Genomics Viewer[28] shows that ERV316A3_2q33.1 g overlaps with the

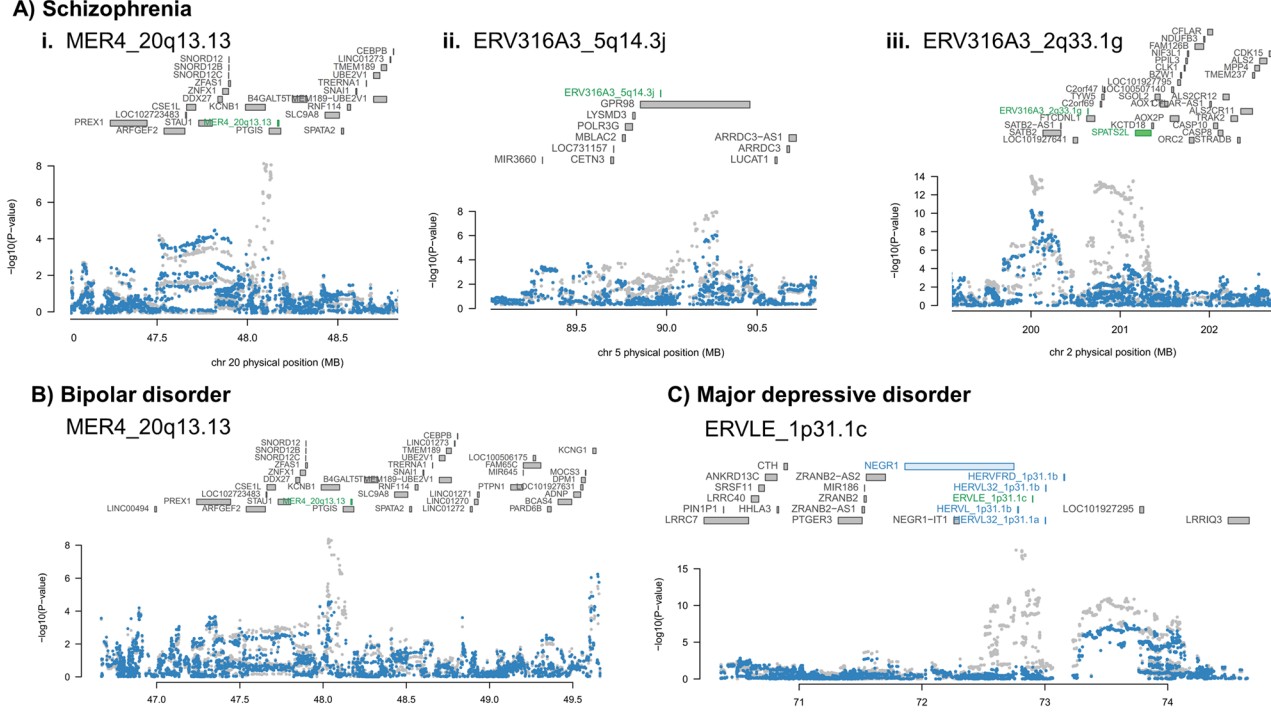

**Fig. 3 | Predicted HERV expression signatures explaining GWAS signals at multiple locations. A** For schizophrenia, we observed instances where the HERV expression signal was the best feature to explain some of the GWAS signal at the locus, e.g., **i.** MER4_20q13.13 and **ii.** ERV316A3_5q14.3j, and an instance where more than one expression feature, including a HERV, were associated with risk, e.g., **iii.** ERV316A3_2q33.1 g. **B** For bipolar disorder, we also observed the expression of MER4_20q13.13 as a feature explaining the GWAS signal at its locus. **C** For major depressive disorder, multiple expression signatures correlated with risk on chromosome 1p31 (feature names labelled in blue), but ERVLE_1p31.1c showed independent association with the disorder (feature name labelled in green). Upper part of each image: genomic context. Lower part of each image: a plot in which the X-axis indicates genomic location, and the Y-axis shows -log10(P) of genetic variant associations (from the GWAS, two-sided), before (grey dots) and after (blue dots) conditioning on jointly significant genes in each locus. *P*-values are not adjusted for multiple testing. Only high confidence risk HERVs are shown.

3' untranslated region of a *FTCDNL1* transcript (Fig. 5C), and that ERV316A3_5q14.3j is in the promoter region of an *ADGRV1* transcript (synonym: *GPR98*; Fig. 5D). These findings suggest that their respective rTWAS signals do likely reflect specific isoforms of these genes, highlighting the importance HERV retrotransposition may have played in the diversification and evolution of gene expression in the modern human genome. On the other hand, MER4_20q13.13 is encoded in the opposite strand of the gene *PTGIS* (Fig. 5E), and ERVLE_1p31.1c is considered intergenic (closest gene is *NEGR1*). We hypothesise that these HERVs likely reflect the existence of non-coding RNAs (ncRNAs) in these regions. This is further supported by the fact that certain ncRNAs have been annotated near ERV-LE_1p31.1c (Fig. 5F). An analysis with Pfam[29] did not identify known protein motifs within these HERV sequences, although further functional studies are necessary to confirm or rule out the production of small proteins by these sequences.

We further explored the genomic context of the high confidence risk HERVs using the UCSC Browser[30], which revealed there are predicted distal enhancers within the predicted locations of ERVLE_1p31.1c (ENCODE accession: EH38E1358923), ERV316A3_2q33.1 g (ENCODE accession: EH38E2064906), and MER4_20q13.13 (ENCODE accession codes: EH38E2118754, EH38E2118755, EH38E2118756, EH38E2118757, EH38E2118758), but none around ERV316A3_5q14.3j. The general abundance of these potential regulatory sequences aligns with the recognised regulatory role attributed to DNA sequences derived from HERVs. However, interpreting their meaning, especially concerning HERV expression, poses challenges. This difficulty arises from the absence of long-read RNA sequencing data that would enable the comprehensive definition of HERV transcripts and their exact genomic positions. Furthermore, predictions of enhancers require experimental validation.

**Co-expression network analysis.** To further investigate the function of HERVs expressed in the DLPFC of 563 samples from individuals of European ancestry, we analysed the expression data using a weighted correlation network analysis (WGCNA)[31]. This analysis was performed based on the premise that genes within expression modules are more likely to share a similar function[32]. We observed 16 expression modules (and an additional 'grey' module containing genes/HERVs that could not be attributed to the expression modules detected; Supplementary Fig. 1). We found that all co-expression modules contained some HERVs (Supplementary Data 4), suggesting a potential role for HERVs in diverse biological functions, although their distribution varied substantially across modules (Fig. 6A). Gene ontology (GO) analysis of the canonical genes belonging to each module identified GO terms ranging from 'synapse' for the 'cyan' module, 'mitochondria' for the 'blue' module, and 'immune response' for the 'greenyellow' module. The top GO term identified per module is shown in Fig. 6B and the top ten Bonferroni significant GO terms per module (Bonferroni $P < 0.05$) are shown in Supplementary Data 5.

The most HERV-enriched and largest detected module was the 'turquoise' module, which comprised of 1398 canonical genes (27% of the module) and 3,815 HERVs (73%), including all four high confidence risk HERVs from Table 2. This module was enriched for GO terms related to signal transduction, such as *G protein-coupled receptor activity* ($P = 1.92 \times 10^{-14}$, Bonferroni $P = 7.05 \times 10^{-9}$) and *detection of chemical stimulus* ($P = 1.64 \times 10^{-21}$, Bonferroni $P = 6.03 \times 10^{-16}$). We tried to force split this large module further by fine-tuning the WGCNA

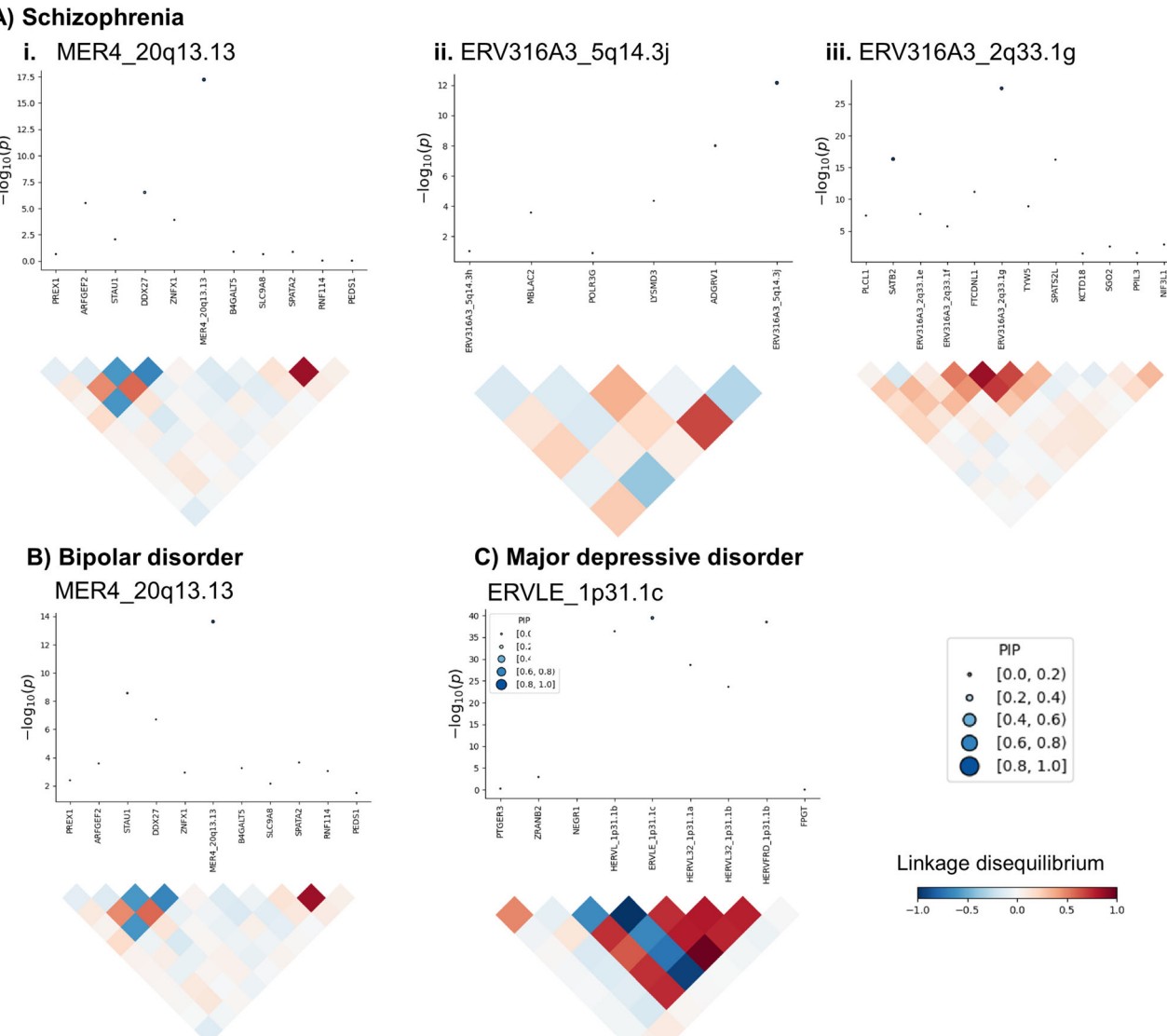

**Fig. 4 | Fine-mapping analysis supports high confidence risk HERVs for multiple psychiatric disorders.** The graphs correspond to the HERV expression signals in the fine-mapping analysis that are also significant in the conditional analyses, in relation to (**A**) schizophrenia, including **i.** MER4_20q13.13, **ii.** ERV316A3_5q14.3j, and **iii.** ERV316A3_2q33.1 g; (**B**) bipolar disorder, which includes MER4_20q13.13; and (**C**) major depressive disorder, which includes ERVLE_1p31.1c. Upper part of each image: graph where the Y-axis indicates the TWAS association *p* value (two-sided), unadjusted for multiple testing, and the X-axis shows genetic features in the linkage disequilibrium block. The size and colour of the points indicate the posterior inclusion probability (PIP), indicating the probability that the expression feature is causal for the association signal at the locus. Lower part: correlation of predicted expression.

**Table 2 | Association statistics pertaining to high confidence risk HERVs from across the rTWAS analyses, including the conditional analyses and fine-mapping results (Europeans only)**

| Trait | HERV ID | Z | P | Bonferroni P | Joint Z | Joint P | PIP |
|---|---|---|---|---|---|---|---|
| Schizophrenia | ERV316A3_2q33.1 g | −7.13 | 1.03E−12 | 8.43E−09 | −5.50 | 2.90E−08 | 1.00 |
| Schizophrenia | ERV316A3_5q14.3j | −4.55 | 5.46E−06 | 0.045 | −4.50 | 5.50E−06 | 0.98 |
| Schizophrenia | MER4_20q13.13 | 5.73 | 9.95E−09 | 8.15E−05 | 5.70 | 1.00E−08 | 1.00 |
| Bipolar disorder | MER4_20q13.13 | 5.04 | 4.73E−07 | 0.004 | 5.00 | 4.70E−07 | 0.99 |
| Major depressive disorder | ERVLE_1p31.1c | 8.72 | 2.91E−18 | 2.35E−14 | 8.70 | 2.90E−18 | 0.68 |

arguments *deepSplit* and *mergeCutHeight*, but all attempts resulted in similar findings, where most genetic features in that module were HERVs, and the overall GO terms assigned to it were generally related to signal transduction. We also ran a parallel analysis after adjusting the expression data for the institution of sample origin, RNA integrity number, sex, case-control status, *post-mortem* interval, age bins, population covariates and surrogate variables, which resulted in similar findings. Finally, a parallel analysis of the 229 samples of African ancestry further provided support for the association between HERVs from the turquoise module and canonical genes linked to signal

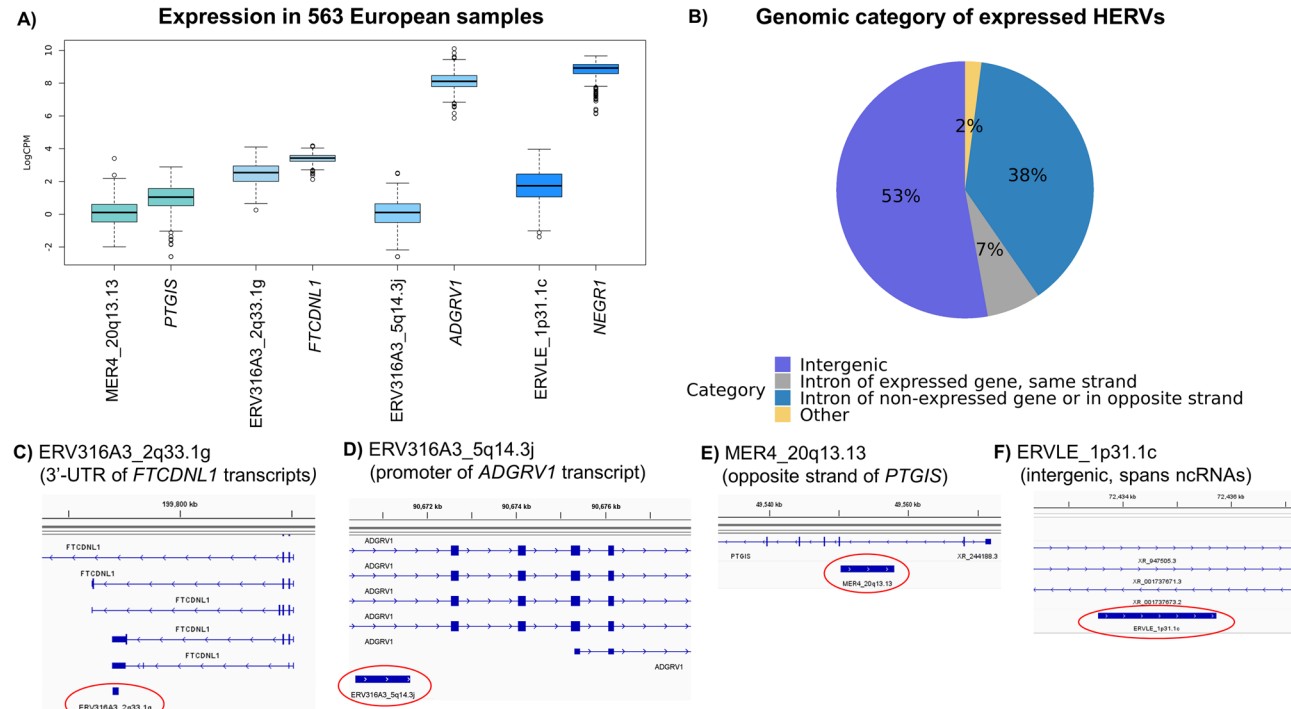

**Fig. 5 | Genomic context of high confidence risk HERVs. A** Expression of HERVs and their nearest canonical genes are shown as median values with interquartile range, with outliers depicted separately (*N* = 563 biologically independent samples of European ancestry). **B** Analysis using HOMER indicates that approximately 98% of HERVs from Telescope are in intergenic and intronic regions, whereas the remainder ('Other') is located in promoters, untranslated regions, or transcription start or termination sites. **C** The genomic context of ERV316A3_2q33.1 g and (**D**) ERV316A3_5q14.3j suggests that these HERVs are likely part of specific isoforms of canonical genes *FTCDNL1* and *ADGRV1*, respectively. On the other hand, (**E**) MER4_20q13.13 is encoded in the opposite strand of the canonical gene *PTGIS*, and (**F**) ERVLE_1p31.1c is intergenic (nearest gene is *NERG1*), suggesting that they are likely producing novel non-coding RNAs.

transduction. In this analysis, the turquoise module comprised of 3678 (71%) HERVs and 1524 (29%) canonical genes, and the same GO terms that were significant in Europeans were also amongst the top ten significant terms in this analysis, e.g., *G protein-coupled receptor activity* ($P = 7.95 \times 10^{-11}$, Bonferroni $P = 2.92 \times 10^{-5}$) and *detection of chemical stimulus* ($P = 6.88 \times 10^{-14}$, Bonferroni $P = 2.52 \times 10^{-8}$).

## Discussion

HERVs have previously been implicated in psychiatric conditions[15–20], but research has been hampered by methodological limitations, small sample sizes and, ultimately, inconsistent findings. In our study, we used a TWAS approach to perform retrotranscriptome imputation for five major psychiatric disorders, using RNA-sequencing and genetic data obtained from samples from a large cohort. This approach employed a specialised bioinformatic tool, Telescope, to quantify HERV expression in RNA-seq data to precise source chromosomal locations[14]. Through integration with GWAS summary statistics, we were able to investigate HERV expression signatures associated with major psychiatric conditions.

We observed two high confidence expression signatures specifically associated with schizophrenia risk (ERV316A3_2q33.1g, ERV316A3_5q14.3j), one shared between schizophrenia and bipolar disorder (MER4_20q13.13), and one associated with major depressive disorder (ERVLE_1p31.1c). We observed that the families to which these HERVs belong (as denoted in their name prefix) are different from those previously highlighted in association with schizophrenia (e.g., HERV-W[19], HERV-K10[20]), bipolar disorder (e.g., HML-2[33]), or major depressive disorder (e.g., HERV-W[17]). The probable reason for such discrepancy is that earlier studies employed HERV quantification methods that averaged expression signals from across multiple HERV copies in the genome (as discussed in the Introduction). In

addition, because these studies aimed to detect case-control differences in samples originating from small cohorts, they were more likely to detect secondary disease expression signatures, including those associated with effects of medication or smoking. Our work, on the other hand, used a specialised HERV expression quantification approach that infers HERV expression levels with genomic precision. It also focuses on expression signatures associated with genetic risk and thus mechanisms more likely to be implicated in disorder aetiology. Considering that specific HERVs from different families were detected in our study in association with psychiatric disorder risk, future studies should also consider HERV expression with genomic precision (instead of simply grouping expression information from within family copies). HERV family assignment is related to the evolutionary trajectory of these sequences within the genome, and it seems like an important parameter for future research. However, we hypothesise that local chromatin modifications and genetic and epigenetic mutations have likely caused different HERVs (even copies from within families) to diverge and exert different roles. Although the high confidence risk HERVs belonged to a large co-expression module comprising of thousands of HERVs, only a selected few are regulated in association with psychiatric disorder risk.

It is not clear yet how the expression of the high confidence risk HERVs may play a role in psychiatric disorders. It was previously hypothesised that differential HERV expression in psychiatric cases was likely to be a by-product of immune responses against current or past infections[34]. Indeed, HERV expression is modulated by exposure to several pathogens[35,36] and can activate inflammatory cascades[37]. This is an interesting theory that corroborates the fact that individuals with psychiatric disorders typically have higher incidences of infections[38–40]. However, our main analysis found that 1238 HERVs

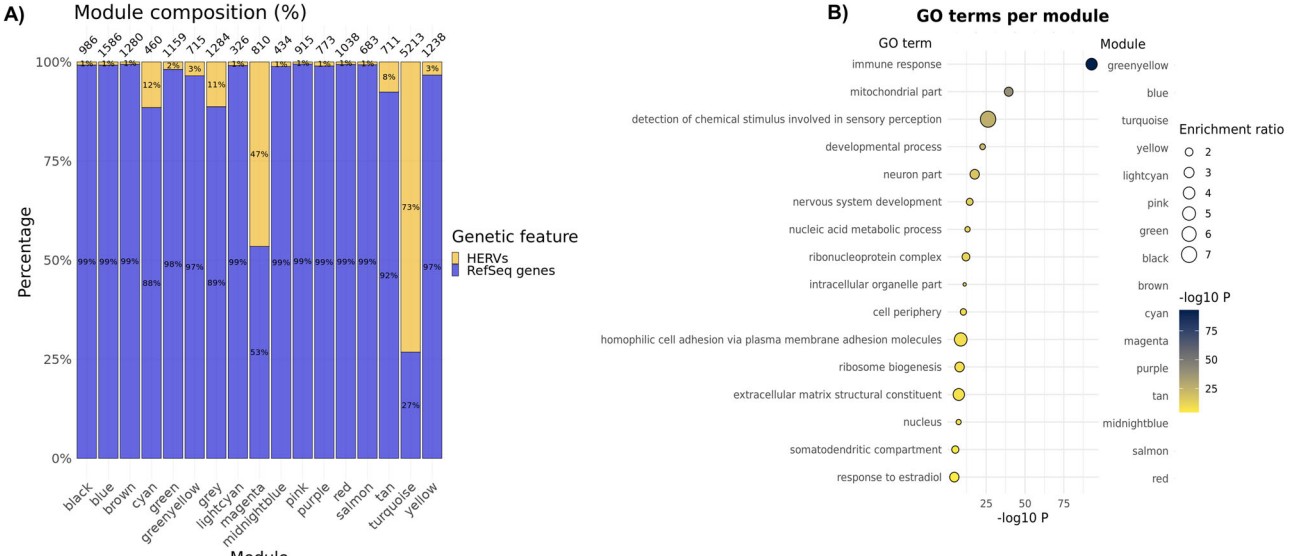

**Fig. 6 | Co-expression analysis identifies HERVs co-expressed with canonical genes and supports their role in a range of biological functions. A** Proportion of HERVs and canonical genes assigned to each co-expression module, including the number of genetic features per module at the top, as detected in the European subset (*N* = 563 biologically independent samples). **B** Bubble plot showing top gene ontology (GO) term, per module. The X-axis and colour of the bubbles indicate −log10(P) (two-sided, uncorrected) of the enrichment statistic. The size of the bubbles represents the enrichment ratio. Only Bonferroni-significant GO terms are shown (Bonferroni-adjusted *P* < 0.05).

expressed in the brain are regulated in *cis*, some of which in association with risk for complex psychiatric traits. This indicates that there are HERV expression mechanisms directly contributing to disorder aetiology, that are not simply part of compensatory responses, or triggered by environmental factors.

In our analysis of European samples, we found 4594 HERVs expressed in the brain, many of which were coregulated with genes playing specialised neurobiological roles, according to a co-expression analysis. While it remains unclear the role specific HERVs play in relation to the GO terms identified, some signalling cascades have been proposed to explain the effects HERV expression exerts on biology. For example, HERV expression activation has been hypothesized to promote the formation of double stranded RNA (dsRNA), which can activate antisense RNA (asRNA) pathways to target gene expression regulation. It can also activate dsRNA-induced signalling pathways and stimulate the production of inflammatory molecules, such as tumour necrosis factor α (TNFα) and interleukin 6 (IL-6), which are known to modulate neuroinflammation[37]. There are other cascades that moderate HERV function, for example those involving Toll-like receptors[41]. Some HERV sequences may also encode regulatory RNAs or proteins that regulate in *trans* the expression of other genes[9]. Ultimately, however, a better understanding of the role specific HERVs play in relation to neurobiology and neuropathology is dependent on the functional characterisation of specific sequences, using relevant models.

There are limitations to our study that must be acknowledged. First, our study explored HERV expression associations with psychiatric disorders in a brain area relevant to psychiatry, the DLPFC[42–44]. However, rTWASs incorporating HERV expression data from additional brain areas, developmental time points and tissues, are likely to reveal additional insights. Second, our rTWAS approach assesses only the *cis*-genetic component of expression, and future studies should investigate HERVs modulated by *trans*-regulatory effects associated with psychiatric disorder susceptibility, as well as the *trans* effects of HERV expression. Third, we used WGCNA to provide insights into biological processes associated with HERVs expressed in the brain. However, there remains a gap between these associations and their true function, and future functional studies should investigate, for

example, how specific HERVs influence cell biology, gene expression regulation, and neuronal electrophysiology, in relation to psychiatric disorder risk, as is currently being done for canonical genes[45–51]. Fourth, the HERVs analysed here are those annotated in the human reference genome, and only whole-genome sequencing of large cohorts (e.g., ref. 11.) will identify nonreference HERVs involved in psychopathology. Fifth, it is plausible that some HERV expression signals detected by Telescope are tagging uncharacterised transcripts of local canonical genes, as discussed above. This is less likely to be true for HERVs like MER4_20q13.13 and ERVLE_1p31.1c, which are in the opposite strand of the canonical genes. The existence of canonical transcripts containing unique HERV sequences that confer increased susceptibility to a psychiatric disorder, however, highlights the importance HERVs played in the diversification and evolution of gene expression in the human genome, as well as their contribution to susceptibility to complex disorders. While it is possible to identify chimeric HERV transcripts using short-read RNA-sequencing, long-read RNA-sequencing studies are likely to be better equipped to identify transcripts originating from repetitive sequences. Ultimately, our work investigating HERV expression with single locus resolution highlights extensive HERV expression and regulation in the adult brain, and further reveals a role for HERVs in psychiatric disorder aetiology.

## Methods

### The CommonMind Consortium dataset

We analysed the CommonMind Consortium dataset to investigate HERV expression mechanisms in the human dorsolateral prefrontal cortex (DLPFC). Access to these data was granted under a Material Transfer Agreement with the National Institute of Mental Health (NIMH) Repository and Genomics Resources (NRGR). Informed consent and permission to share the data had been previously obtained, in compliance with the guidelines specified by the institutional review boards of each recruiting centre involved in sample collection. The *post-mortem* samples were obtained as described in ref. 52. and ref. 53. The initial cohort consisted of 910 distinct individuals from whom expression, genotype, and clinical data were available, as part of the first and third CMC data releases (CMC1 and CMC3, respectively). This sample consisted of individuals that had no psychiatric diagnosis at the

time of death (N = 442), as well as individuals who were diagnosed with schizophrenia (N = 350), bipolar disorder (N = 110), or broadly with an affective disorder (N = 8). In total, 47 individuals were 90+ years old at the time of death (their definite age is omitted for compliance with the Health Insurance Portability and Accountability Act), and the remainder (N = 863) were on average 56.61 years old at the time of death (standard deviation (SD) = 18.90; range = 17−90). This cohort consisted of 337 females (37%) and 573 males (63%). Self-reported ancestries consisted of 621 Europeans (68.2%), 243 Africans (26.7%), 33 Hispanics (3.6%), 12 Asians (1.3%), and 1 other (0.1%). The mean *post-mortem* interval was 22.53 h (SD = 15.39, range = 1.40−168) and the mean RNA integrity number was 7.60 (SD = 0.89, range = 4.50−9.60). Total RNA was extracted from autopsy tissue using the RNeasy kit (QIAGEN, Hilden, Germany). For CMC1, ribosomal RNA was depleted using the Ribo-Zero Magnetic Gold kit (Illumina, San Diego, California, United States), and libraries constructed using the TruSeq RNA Sample Preparation Kit v2 (Illumina). For CMC3, ribosomal RNA was depleted using the KAPA RiboErase protocol (F. Hoffmann-La Roche, Basel, Switzerland), and libraries were constructed using the KAPA Stranded RNA-seq Kit. There is currently no information on the polyadenylation status of HERVs, and thus a total RNA sequencing approach followed by ribosomal depletion seems adequate to ensure the capture of HERV expression signals, particularly as they are likely to encode non-coding RNAs, which are typically not polyadenylated[54−57]. The libraries were sequenced on a HiSeq 2500 (Illumina). DNA was extracted using the DNeasy Blood and Tissue Kit (QIAGEN) according to the manufacturer's protocol. For whole-genome genotyping, the CMC1 samples were genotyped using the Infinium HumanOmniExpressExome 8 1.1b chip (Illumina), and the CMC3 samples were genotyped using the Illumina HumanHap650Y, Human1M-Duo, or HumanOmni5M-Quad chips, as described by the authors[52,53].

## Whole-genome genotype data processing

Genotype files based on the genome builds hg19 (CMC1) or hg38 (CMC3) were downloaded[52,53] and formatted using PLINK 1.9[58] and bcftools 1.9[59]. Imputation took place within the Michigan Imputation Server 1.7.4[60], where variants were lifted to hg19, for compatibility with GWAS summary statistics. Imputation was performed for each chip separately using Eagle v2.4 phasing and the 1000 Genomes Phase 3 v5 (mixed population) as reference panel. We analysed only non-ambiguous autosomal single nucleotide polymorphisms (SNPs) with minor allele frequency >0.05, Hardy-Weinberg $P < 5 \times 10^{-6}$, and missing genotype rates <0.05. We removed samples with excess heterozygosity (mean heterozygosity rate above 3 standard deviations), high likelihood of relatedness (pihat > 0.2), those with missing genotype information >0.05, or with mismatched sex information[61].

## Sample selection

We selected individuals of European and African ancestries to construct the TWAS weights, given that these ancestries represented the two largest, most homogenous subsets of the entire sample. To achieve this, the CMC genotype files were analysed using code from the Ancestry_identifier.R script from the GenoPred pipeline[62−64], which uses the 1000 Genomes Phase 3 sample as reference to impute ancestry. We identified 563 individuals of European ancestry, including 242 unaffected individuals, 223 individuals diagnosed with schizophrenia, 91 with bipolar disorder, and 7 broadly diagnosed with an affective disorder. Besides the 27 individuals who were >90 years old, the remaining individuals (N = 536) were on average 58.85 years old at the time of death (standard deviation (SD) = 18.78; range = 17−90). The cohort consisted of 196 females (35%) and 367 males (65%). The mean post-mortem interval was 20.99 h (SD = 12.87, range = 2.00−84.50) and the mean RNA integrity number was 7.60 (SD = 0.91, range = 4.60−9.60). We included individuals with a psychiatric diagnosis in the construction of the SNP weights, as the added sample size increases

power to detect *cis*-regulatory effects associated with GWAS traits, as demonstrated in the sensitivity tests described in the Results. We also identified 229 individuals of African ancestry, which consisted of 139 unaffected individuals, 80 individuals diagnosed with schizophrenia, 9 with bipolar disorder, and 1 broadly diagnosed with an affective disorder. Besides the 7 individuals who were >90 years old, the remaining individuals (N = 222) were on average 49.45 years old at the time of death (standard deviation (SD) = 17.41; range = 17−89). The cohort consisted of 93 females (41%) and 136 males (59%). The mean post-mortem interval was 28.04 h (SD = 12.67, range = 1.60−168.00) and the mean RNA integrity number was 7.65 (SD = 0.88, range = 5.60−9.30).

## RNA-sequencing data processing

For CMC1 files, we downloaded bam files containing mapped and unmapped RNA-seq reads, and merged and processed them using samtools 1.5[65] and the flag '-F 0x100' to obtain FASTQ files. For CMC3, we extracted FASTQ files using the SamToFastq function from Picard 3.1.1[66]. We used Trimmomatic 0.38[67] to prune low quality bases (leading/trailing sequences with phred score <3, or those with average score <15 every four bases), or reads below 36 bases in length. For HERV expression quantification, we mapped trimmed reads to the human genome hg38 using Bowtie2 2.3.5.1[68] and the parameters '--very-sensitive-local --k 100 --score-min L,0,1.6'. Subsequently, we used Telescope 1.0.2 to quantify HERV expression using the HERV annotation v2 (hg38) (https://github.com/mlbendall/telescope_annotation_db)[14]. Telescope quantifies HERV expression with genomic precision by reassigning ambiguously mapped reads to the most probable source transcript as determined within a Bayesian statistical model, based on an expectation-maximisation algorithm. This approach diverges from that of other transposable element quantification software, such as ERVmap[69], which opts to discard reads containing mismatches rather than attempting to identify their most likely chromosomal source[70]. In particular, the HERVs Telescope investigates comprise putative transcriptional units containing an internal protein-coding region flanked by LTR regulatory regions[14]. For the quantification of canonical genes, trimmed reads were pseudoaligned to the human reference genome hg38 using kallisto 0.44.0[71]. In R 3.6.3 (The R Project for Statistical Computing, Vienna, Austria), we used tximport 1.14.0[72] to import the kallisto files using the function 'countsFromAbundance = "lengthScaledTPM"', and biomaRt 2.42.0[73] to select canonical genes. In our study, canonical genes, which are predominantly protein coding, were defined based on the presence of a gene symbol established by the HUGO Gene Nomenclature Committee. We combined the expression data pertaining to canonical genes and HERVs and considered "expressed" those features with read counts ≥ 6 and transcripts per million (TPM) ≥ 0.1 in at least 20% of samples, in accordance with the GTEx Consortium guidelines for processing RNA-seq data for eQTL analysis[74]. Principal component analysis was employed for visual inspection to identify and subsequently remove obvious outliers. The HERV and gene coordinates were lifted to hg19 using liftOver[75] for compatibility with the GWAS summary statistics. We explored genetic categories represented within the HERV annotation using HOMER 4.11[27].

## Summary statistics

Summary statistics from the European subset of the latest schizophrenia GWAS, performed by ref. 23. were downloaded from the Psychiatric Genomics Consortium (PGC) website. We also downloaded summary statistics corresponding to GWASs of bipolar disorder[76], major depressive disorder (except 23andMe)[77], attention deficit hyperactivity disorder[78], and autism spectrum conditions[79] in Europeans. To explore the translatability of our findings to different ancestries, we performed cross-ancestry validation analyses using schizophrenia GWAS summary statistics from African American,

Latino, and Asian cohorts[23], and a major depressive disorder GWAS summary statistics from an East Asian cohort[24]. We analysed only biallelic non-ambiguous single nucleotide polymorphisms with imputed minor allele frequency >5% (calculated based on the European subset of the 1000 Genomes reference panel), and imputation score >0.80.

## rTWAS

Following the quantification of canonical genes and HERVs using kallisto and Telescope, respectively, we integrated these data and normalised them using the trimmed mean of M values (TMM) method separately for the European and African samples[80]. For each subset, we created SNP weights that combine expression data from males and females, based on the assumption that *cis*-heritable expression is mostly shared across sexes[81], and on the fact that the combined sample provides additional power to detect genetic features with *cis*-heritable expression. We used limma 3.42.0[82] to adjust the expression data for the institution of sample origin, case-control status, RNA integrity number, sex, post-mortem interval, age (determined in bins: #1 = 17−29 years, #2 = 30−49 years, #3 = 50−69 years, #4 = 70−89 years, #5 = 90+ years), the first ten population covariates estimated through a principal component analysis performed in PLINK 1.9[58], and surrogate variables calculated using sva 3.34.0[83], following previous work[53,84]. The number of surrogate variables was determined as a function of sample size (N), as suggested by GTEx (i.e., 30 for sample sizes between 150 and 250, and 60 for sample sizes above 350). We adapted scripts from https://github.com/opain/Calculating-FUSION-TWAS-weights-pipeline[22,85] to construct FUSION SNP weights. Briefly, this process used the FUSION.compute_weights.R script[86] to estimate *cis*-heritable genes in the expression data originating from the European or African CMC subset within 1 Mb windows, using the European or African subset of the 1000 Genomes Phase 3 as reference population, respectively. We calculated the SNP weights for each ancestry using the methods blup (Best Linear Unbiased Predictor computed from all SNPs), bslmm (Bayesian Sparse Linear Model), lasso (lasso regression), elastic net (Elastic-net regression), and top SNP (single best expression quantitative trait locus), except for instances where blup or bslmm were excluded due to convergence issues. The rTWASs were performed through analysis of GWAS summary statistics using FUSION[21] and our customised SNP weights, controlling for linkage disequilibrium using genetic data from the CMC subset that was used to create the weights. We matched ancestry from SNP weights and GWAS results for the rTWASs, unless stated otherwise (e.g., in cross-ancestry analyses, where SNP weights and the LD reference panel were of European ancestry, but the GWAS results were from a non-European ancestry). We applied multiple testing correction to the rTWAS association signals per trait using the Bonferroni method, considering the total number of tested genetic features. Plots were generated and analyses performed using the FUSION pipeline and scripts adapted from https://opain.github.io/MDD-TWAS/[5] and https://github.com/rodrigoduarte88/hiv-meta-twas-2021[87].

## rTWAS secondary analyses

We performed sensitivity analyses to test whether HERV expression signals were able to explain GWAS signals, competitively against canonical genes. To achieve this, we performed conditional analyses using FUSION[21] to estimate the proportion of the GWAS signals that were explained by rTWAS signals within each loci. We also performed fine-mapping analyses using FOCUS[88] to identify the strongest expression association signal within each linkage disequilibrium block after controlling for the correlation of neighbouring signals. FOCUS calculates the posterior inclusion probability (PIP) for each expression signature in an LD block to be causal given the observed rTWAS statistics, whereby those with PIP > 0.50 are more likely to be causal than other features at the locus.

We performed additional analyses to explore the translatability of findings obtained in Europeans to other populations. First, we performed rTWASs using SNP weights calculated in Europeans with GWAS results obtained from analysis of individuals from different ancestries[23,24]. Cross-ancestry validation would be considered significant if the signal identified in Europeans was also identified in other populations in the same direction of effect and if it survived multiple testing correction for the number of expression signatures tested in the rTWAS (Bonferroni $P < 0.05$). Second, since we constructed SNP weights using the African American subset of the CMC, we also performed an rTWAS of schizophrenia in African Americans using GWAS results obtained from the African American subset of the PGC's schizophrenia study[23].

## Weighted Correlation Network Analysis (WGCNA)

We used WGCNA 1.69 to identify co-expressed genes and HERVs in the RNA-seq data, in order to infer the biological function of expressed HERVs[31]. WGCNA is a powerful systems biology method that has been previously used to predict the biological function of uncharacterised genes and non-coding RNAs[89,90]. We constructed a signed expression network consisting of HERVs and canonical genes expressed in the DLPFC. The expression data was TMM-normalised and used to create an adjacency matrix to inform the co-expression similarity observed between all pairs of expressed genes and HERVs (i.e., genes and genes, genes and HERVs, HERVs and HERVs). Module identification was performed by applying hierarchical clustering to the adjacency matrix of expression data, filtering spurious relationships through the application of a topological overlap approach. We used an $R^2$ cut-off of ~0.8, which corresponded to a $\beta = 12$, to construct the network. Each module was arbitrarily assigned a colour, and genes or HERVs not belonging to any module were assigned to the grey module.

## Gene Ontology (GO) analyses

We performed GO analyses in R using anRichment 1.22[91] to identify the function of the modules classified through WGCNA. This was performed to infer the potential function of HERVs expressed in the cohort subset, based on the function of canonical genes belonging to each module. We used the Bonferroni method to correct for multiple testing (Bonferroni $P < 0.05$). The gene ontology plot was created using code adapted from ref. 92.

## Statistical analyses

Analyses were performed using King's College London's High Performance Computing Cluster CREATE[93], in Bash 5.0.17 (GNU Project Bourne Again SHell) and R 3.6.3 (The R Project for Statistical Computing, Vienna, Austria). Correlations were calculated in R using the cor.test() function.

## Reporting summary

Further information on research design is available in the Nature Portfolio Reporting Summary linked to this article.

# Data availability

The RNA-sequencing and genotype data from the CommonMind Consortium cohort are available under restricted access for containing sensitive data. Access can be obtained via an application to the NIMH Repository and Genomics Resource (NRGR https://www.synapse.org/#!Synapse:syn2759792/). GWAS summary statistics were downloaded from the Psychiatric Genomics Consortium website (https://pgc.unc.edu/for-researchers/download-results/). SNP weights derived from our analyses and example reference panels are freely available from King's College London Research Data Repository (KORDS) (https://doi.org/10.18742/22179655)[94]. All other data generated during this study are included in this published article and its supplementary information files.

## Code availability

All code used in the manuscript is available from King's College London Research Data Repository (KORDS) (https://doi.org/10.18742/22179655)[94] and GitHub (https://github.com/rodrigoduarte88/TWAS_HERVs-SCZ). A tutorial to perform an rTWAS is available at https://rodrigoduarte88.github.io/neuro_rTWAS.

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

## Acknowledgements

Research reported in this publication was supported by the National Institutes of Health (NIH) under award number R21 HG011513 to T.R.P., D.F.N. and R.R.R.D. The content is solely the responsibility of the authors and does not necessarily represent the official views of the NIH. T.R.P. is supported by an MRC (UKRI) New Investigator Research Grant (MR/W028018/1). For the purpose of open access, the author has applied a CC BY public copyright license to any Author Accepted Manuscript version arising from this submission. This research is also part-funded by the National Institute for Health and Care Research (NIHR) Maudsley Biomedical Research Centre at South London and Maudsley National Health Service (NHS) Foundation Trust and King's College London. The views expressed are those of the authors and not necessarily those of the NHS, the NIHR or the Department of Health and Social Care. Data for this publication were obtained from the National Institute of Mental Health (NIMH) Repository & Genomics Resource, a centralised national biorepository for genetic studies of psychiatric disorders. The data were generated as part of the CommonMind Consortium, supported by funding from Takeda Pharmaceuticals Company Limited, F. Hoffman-La Roche Ltd and NIH grants R01MH085542, R01MH093725, P50MH066392, P50MH080405, R01MH097276, RO1-MH-075916, P50M096891, P50MH084053S1, R37MH057881, AG02219, AG05138, MH06692, R01MH110921, R01MH109677, R01MH109897, U01MH103392, and contract HHSN271201300031C through IRP NIMH. Brain tissue for the study was obtained from the Mount Sinai NIH Brain and Tissue Repository, the University of Pennsylvania Alzheimer's Disease Core Center, the University of Pittsburgh NeuroBioBank and Brain and Tissue Repositories, and the NIMH Human Brain Collection Core. Thanks to CMC Leadership, including Panos Roussos, Joseph Buxbaum, Andrew Chess, Schahram Akbarian, Vahram Haroutunian (Icahn School of Medicine at Mount Sinai), Bernie Devlin, David Lewis (University of Pittsburgh), Raquel Gur, Chang-Gyu Hahn (University of Pennsylvania), Enrico Domenici (University of Trento), Mette A. Peters, Solveig Sieberts (Sage Bionetworks), Thomas Lehner, Stefano Marenco, Barbara K. Lipska (NIMH).

## Author contributions

Study design and conception: R.R.R.D., T.R.P., D.F.N. Performed analyses: R.R.R.D. Statistical support: O.P. Wrote the paper: R.R.R.D., T.R.P. Intellectual input, revised the manuscript: O.P., M.L.B., M.M.R., J.L.M., S.S., C.T., S.K.L., R.A.B., J.M., P.F.O., D.P.S., D.F.N.

## Competing interests

The authors declare no competing interests.

## Additional information

¹Social, Genetic and Developmental Psychiatry Centre, Institute of Psychiatry, Psychology & Neuroscience, King's College London, London, UK. ²Division of Infectious Diseases, Weill Cornell Medicine, Cornell University, New York, NY, USA. ³Department of Basic and Clinical Neuroscience, Institute of Psychiatry, Psychology and Neuroscience, King's College London, London, UK. ⁴MRC London Neurodegenerative Diseases Brain Bank, Institute of Psychiatry, Psychology and Neuroscience, King's College London, London, UK. ⁵Department of Clinical and Biomedical Sciences, University of Exeter Medical School, University of Exeter, Exeter, UK. ⁶Department of Genetics and Genomic Sciences, Icahn School of Medicine, Mount Sinai, New York, NY, USA. ⁷MRC Centre for Neurodevelopmental Disorders, King's College London, London, UK. ⁸Feinstein Institutes for Medical Research, Northwell Health, Manhasset, NY, USA. ⁹These authors jointly supervised this work: Douglas F. Nixon, Timothy R. Powell. ✉e-mail: rodrigo.duarte@kcl.ac.uk; timothy.1.powell@kcl.ac.uk

