## [Peer Review File · Nature Communications]

Integrating human endogenous retroviruses into transcriptome-wide association studies highlights novel risk factors for major psychiatric conditionsEditorial Note: Parts of this Peer Review File have been redacted as indicated to maintain the confidentiality of unpublished data.

Reviewer #1 (Remarks to the Author):

In this new study, R. Duarte and colleagues explored the human endogenous retrovirus (HERV) transcription activity in the dorsolateral prefrontal cortex and their potential impacts on genetic susceptibility to five major psychiatric conditions (schizophrenia, bipolar disorder, autism, depressive disorder, and attention deficit). The authors developed and performed a transcription-wide association study (TWAS) focused on HERV activity (= retrotranscriptome-wide association studies (rTWAS)) using more than 500 brain samples from the CommonMind Consortium (CMC). The authors found a strong transcriptional activity of HERV elements (> 1300 elements) in brain, and they were able to associate a low number of them with psychiatric conditions. I would like to point out the huge number of datasets that the authors analyzed and the idea of incorporating transcription element (TE) activity into the GWAS or TWAS analysis is interesting, and it will be a nice contribution to the literature.

However, I have major reservation in the RNA-seq analysis of HERV elements. The authors used Telescope, a TE-dedicated software to access HERV transcription, which is an accurate tool to prevent misinterpretation of RNA-seq data on repeated elements. However, because they analyzed total RNA datasets (and not polyA+), I wonder how the analysis is actually accurate to discriminate signal from TE promoter to pervasive transcription or pre-mRNA signal. In fact, given the abundance of TEs in intronic regions, the use of total RNA increases the pervasive signal and not reflect transcription from HERV transcripts. I am wondering if the estimation of expressed HERV loci is overestimated here and if that could have an influence on the rTWAS interpretation?

They authors mention several times that some HERV loci are "coregulated with genes" and I am not sure to understand what it means. Do they consider that some HERV considered as expressed are intronic elements located in an expressed gene and the observed signal is not from HERV promoters?

Besides, I think it would be interesting to go further on HERV analysis. Are they able to associate the expression of the HERV locus with its integrity? The elements found as expressed are they all full-length, or with a functional promoter, presence of cis regulatory sequences? What are the mechanisms that could explain that a HERV element could represent a genetic risk in the dorsolateral prefrontal cortex? For example, are the authors able to associate the presence of a HERV element with chimeric transcripts? Three elements showed in the figure 4 are located at the 3' end of genes. Are these genes expressed? Do these HERV elements "interfere" with the 3' termination of these genes? A better characterization of the "risk HERVs" may be important for future studies.

Finally, I found that the figures are difficult to understand for a non-specialist audience and would require more work to make them accessible.

Best regards.

Reviewer #2 (Remarks to the Author):

Human Endogenous Retrovirus (HERV) have been linked to all major psychiatric conditions but their role in etiology remains elusive and previous research has been limited. Duarte et al developed a specialized approach termed rTWAS to perform retrotranscriptome imputation and association study using available RNA-seq and genetic data from 590 brain samples. Through integration with GWAS summary statistics followed by conditional and fine-mapping analyses, the authors investigated HERV expression signatures associated with five major psychiatric conditions and identified one association each for two of the conditions evaluated. The work is original and the results are noteworthy. Methods are solid and well described most of the time.

1. The authors employed a specialized bioinformatic tool called Telescope that they published previously. The authors only provided a reference to Telescope which is not sufficient. Suggest the authors summarize the methodological details of Telescope in the present manuscript so that the

readers do not have to find the original paper in order to understand kind of what has been done.

2. In the Methods section, the authors stated that SNP weights were constructed using individuals of European ancestry and of both cases and controls. This is not convincing. Can you provide actual evidence, (e.g. by performing a sensitivity analysis), that (1) European SNP weight is appropriate for cross-ancestry samples, and (2) cases and controls can be combined?

3. In the Discussion section, the author reported that the HERV families they identified are different from those previously reported in the literature. Can the authors elaborate more on this inconsistency? What might contribute to it?

4. Related to the comment above, part of the inconsistency could be a result of the still-small sample size of this study. I suggest that the authors perform a statistical power analysis of the rTWAS.

#
# Reviewer #1 (Remarks to the Author):
#
# In this new study, R. Duarted and colleagues explored the human endogenous retrovirus (HERV)
# transcription activity in the dorsolateral prefrontal cortex and their potential impacts on
# genetic susceptibility to five major psychiatric conditions (schizophrenia, bipolar disorder,
# autism, depressive disorder, and attention deficit). The authors developed and performed a
# transcription-wide association study (TWAS) focused on HERV activity (=
# retrotranscriptome-wide association studies (rTWAS)) using more than 500 brain samples from
# the CommonMind Consortium (CMC). The authors found a strong transcriptional activity of HERV
# elements (> 1300 elements) in brain, and they were able to associate a low number of them with
# psychiatric conditions. I would like to point out the huge number of datasets that the authors
# analyzed and the idea of incorporating transcription element (TE) activity into the GWAS or
# TWAS analysis is interesting, and it will be a nice contribution to the literature.
#

Dear Reviewer 1, many thanks for summarising our research and for your kind comments on the innovative
nature and significance of our work.

Please note that as we worked towards this revised submission of our manuscript, a colleague brought to
our attention the presence of samples with mismatched sex information in the CommonMind Consortium
dataset. We subsequently performed sex checks using the genetic data and excluded samples with
mismatched sex information to mitigate the potential introduction of bias into our analyses. Consequently,
some of our results underwent changes, though most high-confidence risk HERVs remained significant in
our new set of results.

#
# However, I have major reservation in the RNA-seq analysis of HERV elements. The authors used
# Telescope, a TE-dedicated software to access HERV transcription, which is an accurate tool to
# prevent misinterpretation of RNA-seq data on repeated elements. However, because they analyzed
# total RNA datasets (and not polyA+), I wonder how the analysis is actually accurate to
# discriminate signal from TE promoter to pervasive transcription or pre-mRNA signal. In fact,
# given the abundance of TEs in intronic regions, the use of total RNA increases the pervasive
# signal and not reflect transcription from HERV transcripts. I am wondering if the estimation of
# expressed HERV loci is overestimated here and if that could have an influence on the rTWAS
# interpretation?
#

Thank you for these comments, we appreciate your attention to this critical aspect of our study. We justify the
choice of the total RNA dataset analysed based on four reasons. **First**, most HERVs are predicted to give
rise to non-coding RNAs, molecules that are typically not polyadenylated (Akrami et al., 2013; Du et al.,
2013; Esteller, 2011; Fatica & Bozzoni, 2014). For instance, Akrami and colleagues (Akrami et al., 2013)
found that out of 25.7 billion GENCODE-mapped read pairs from all polyA+ RNA-seq datasets, only 225
million read pairs (<1%) mapped to lncRNA loci, illustrating how uncommon it is for non-coding RNAs to be
polyadenylated, and emphasizing the need for sequencing methods based on total RNA to accurately
quantify and identify these molecules. To provide a more clear panorama on why this is relevant to our study,
previous work (Stearrett et al., 2021) suggests that about 94% of HERV elements within the Telescope
annotation (i.e., HERV transcriptional units encoding retroviral genes like pol, env, or gag, flanked by viral
promoters known as long terminal repeats) are predicted to encode non-coding RNAs (predictions available
at github.com/liniguez/Telescope_MetaAnnotations/blob/main/TE_annotation.v2.0.tsv). **Second**, many
studies looking at HERV expression using next generation sequencing technology have analysed datasets
originating from total RNA (e.g., Neulinger-Munoz et al. (2021), Guffanti et al. (2018)). **Third**, it is important to
highlight that we were also concerned about pre-mRNA signals potentially biasing HERV expression
quantification in our samples. The way we mitigated this was by applying stringent expression cut-off criteria
to define genes and HERVs as “expressed” in our dataset, which take into consideration degree and
consistency of expression across samples (as defined within the guidelines from GTEx to process RNA-seq
data for eQTL analysis). See our expression cut-off criteria below, as discussed in our *Methods*:

- *“We combined the expression data pertaining to protein-coding genes and HERVs and considered “expressed” those features with read counts ≥ 6 and transcripts per million (TPM) ≥ 0.1 in at least 20% of samples, in accordance with the GTEx Consortium guidelines for processing RNA-seq data for eQTL analysis (GTEx Consortium et al., 2017).”*

Thus, assuming that pre-mRNA signals occur stochastically across samples, it is unlikely that they would be
persistent across this many samples by chance (20% of 563 = at least 112 samples). The expression cut-off
used gives us an indication that the signals identified within the Telescope pipeline are real and robust.

**Fourth**, and also assuming stochastic pre-mRNA signals, we assume that the signal would occur randomly
with respect to genotype, thus significant associations according to genotype would be unlikely to be
observed (which is not the case, as per our rTWAS findings).

We do think that library preparation can have an effect on gene and HERV expression quantification, but in
the absence of expression data generated after Poly-A selection in the CommonMind Consortium samples,
we opted for simply stating our approach in the manuscript, not as a limitation, but as something for readers
to be aware. Thus, we have added this sentence in the Methods:

- *“There is currently no information on the polyadenylation status of HERVs, and thus a total RNA sequencing approach followed by ribosomal depletion seems adequate to ensure the capture of HERV expression signals, particularly as they are likely to encode non-coding RNAs, which are typically not polyadenylated (Akrami et al., 2013; Du et al., 2013; Esteller, 2011; Fatica & Bozzoni, 2014).”*

#

**# They authors mention several times that some HERV loci are “coregulated with genes” and I am**
**# not sure to understand what it means. Do they consider that some HERV considered as expressed**
**# are intronic elements located in an expressed gene and the observed signal is not from HERV**
**# promoters?**

#

Thank you for this comment. To provide some clarification, HERVs can be coregulated with protein coding
genes in two (non-mutually exclusive) ways, as per the results of our study. First, HERVs can be part of co-
expression networks independently from their source genomic locations, which we detected as part of our
analysis using WGCNA. This is because WGCNA identifies co-expressed genetic elements based on co-
expression similarities observed between all pairs of genetic features expressed across samples (Langfelder
& Horvath, 2008). Second, genetic variants regulating HERV expression can also regulate the expression of
multiple neighbouring genetic features because of the effects of linkage disequilibrium, something that
normally occurs for protein coding genes (Gusev et al., 2016), and that we observed for some HERVs (e.g.
HERVs at the 1p31 locus in relation to major depressive disorder risk).

However, this comment may be more concerned with the point raised in the previous comment, regarding
potential pervasive signals arising from unspliced pre-mRNAs, which we addressed in our comment above.
Having justified our choice for analysing a total RNA dataset, and in light of your comments, we would like to
highlight that in the new version of the manuscript, we discuss more clearly the fact that it is possible that
some of the rTWAS signals are picking up expression signals relating to specific isoforms of protein coding
genes (because of isoforms containing HERV sequences). This does not dampen our enthusiasm for our
findings, as this provides evidence of the contribution that HERVs played in the shaping and evolution of our
genome and susceptibility to complex traits. In fact, Sekar et al. (2016) found that a HERV sequence within
the C4 locus is responsible for regulating the ratio of C4A to C4B expression in the brain, which has the
potential to confer risk to schizophrenia by regulating synaptic pruning, a process often considered abnormal
in schizophrenia.

We also discuss in our paper that, on the other hand, some rTWAS signals reflect HERV sequences that are
in the opposite strand of their nearest protein coding genes (e.g. MER4_20q13.13), thus suggesting the
existence of novel non-coding RNAs. As the RNA-seq libraries analysed were stranded, we have a high
degree of certainty that the reads corresponding to those HERVs were correctly mapped, as stranded
libraries provide information about the directionality of transcription in the analysed samples. Furthermore, it

is important to highlight that we found converging evidence from our conditional analyses and fine-mapping
to suggest that expression of HERVs (particularly high confidence risk HERVs, and not neighbouring protein
coding genes) significantly explained the GWAS signals at their respective loci.

[Redacted]

#

**# Besides, I think it would be interesting to go further on HERV analysis. Are they able to**
**# associate the expression of the HERV locus with its integrity? The elements found as expressed**
**# are they all full-length, or with a functional promoter, presence of cis regulatory sequences?**
**# What are the mechanisms that could explain that a HERV element could represent a genetic risk**
**# in the dorsolateral prefrontal cortex? For example, are the authors able to associate the**
**# presence of a HERV element with chimeric transcripts? Three elements showed in the figure 4 are**
**# located at the 3' end of genes. Are these genes expressed? Do these HERV elements "interfere"**
**# with the 3' termination of these genes? A better characterization of the "risk HERVs" may be**
**# important for future studies.**

#

Thank you for this comment. These are excellent points that we are interested to uncover, but we do not
have data to explore whether HERVs of interest may form chimeric transcripts, or if they cause early
termination of neighbouring protein-coding genes. The data we have access to was generated from short
read RNA-sequencing technology, which poses substantial challenges for the identification of transcripts
originating from repetitive elements. In light of your comment, however, we have performed an extensive
manual search on multiple UCSC Browser tracks to identify potential interesting features pertaining to the
high confidence risk HERVs. We have included a brief description of the results of this analysis in the new
version of the manuscript. Briefly, we found putative cis-regulatory sequences within some HERVs, and
included a brief discussion on the challenges in interpreting what this means (e.g., there is no functional
validation of the predicted cis regulatory sequences found, and we also do not fully understand what the
HERV transcripts look like exactly, since transcript detection for repetitive elements using short read RNA-
seq data represents a bioinformatic challenge). We hope that the Reviewer will understand that our efforts to
explore these aspects are ongoing and that further investigations and advancements in HERV research will
contribute to a more comprehensive understanding of their functional implications in future studies.

#

**# Finally, I found that the figures are difficult to understand for a non-specialist audience and**
**# would require more work to make them accessible.**

**# Best regards.**

#

Thank you for your valuable feedback. In response to your comment, we have made significant efforts to
enhance the accessibility of our figures for a non-specialist audience in the revised version of the
manuscript. We have carefully selected more representative images for inclusion in the main article, reduced
the number of unnecessary images, and improved the overall quality and clarity of the visuals. Additionally,
we have renamed the figure legends to ensure that they provide sufficient and detailed explanations for each
image, aiming to make the content more understandable and user-friendly for readers who may not be
specialists in the field. We hope these revisions address your concerns.

#

**# Reviewer #2 (Remarks to the Author):**

**# Human Endogenous Retrovirus (HERV) have been linked to all major psychiatric conditions but**
**# their role in etiology remains elusive and previous research has been limited. Duarte et al**
**# developed a specialized approach termed rTWAS to perform retrotranscriptome imputation and**
**# association study using available RNA-seq and genetic data from 590 brain samples. Through**
**# integration with GWAS summary statistics followed by conditional and fine-mapping analyses, the**
**# authors investigated HERV expression signatures associated with five major psychiatric**
**# conditions and identified one association each for two of the conditions evaluated. The work is**
**# original and the results are noteworthy. Methods are solid and well described most of the time.**

#

Dear Reviewer 2, thank you for summarising our research and for your kind assessment of our work.

#

**# 1. The authors employed a specialized bioinformatic tool called Telescope that they published**
**# previously. The authors only provided a reference to Telescope which is not sufficient. Suggest**
**# the authors summarize the methodological details of Telescope in the present manuscript so that**
**# the readers do not have to find the original paper in order to understand kind of what has been**
**# done.**

#

Thank you for this comment. As per your suggestion, we have added a better description to the Telescope
pipeline in the *Methods*:

- • *“Telescope quantifies HERV expression with genomic precision by reassigning ambiguously mapped*
*reads to the most probable source transcript as determined within a Bayesian statistical model*
*based on an expectation-maximization algorithm. This approach diverges from that of other*
*transposable element quantification softwares, such as ERVmap, which opts to discard reads*
*containing mismatches rather than attempting to identify their most likely chromosomal*
*source(Iñiguez et al., 2019).”*

#

**# 2. In the Methods section, the authors stated that SNP weights were constructed using**
**# individuals of European ancestry and of both cases and controls. This is not convincing. Can**
**# you provide actual evidence, (e.g. by performing a sensitivity analysis), that (1) European SNP**
**# weight is appropriate for cross-ancestry samples, and (2) cases and controls can be combined?**

#

Thank you for these comments. Regarding point (1), about the use of European SNP weights in cross-
ancestry analyses, we understand that employing weights calculated in a sample from one ancestry to
GWAS results corresponding to a different ancestry is not ideal, but it can still lead to informative results
(e.g., Levey et al. (2021)). However, in light of your comment, and in an effort to promote more diverse work
in genomics, we have performed additional analyses, described under a new Results section (*“Risk HERV*
*signatures in non-European ancestries”*, line 398). Briefly, we maintained some of our cross-ancestry tests,
while including the Levey et al citation to demonstrate that this type of work can lead to informative findings.
Additionally, we have calculated new SNP weights using the African American subset of the Common Mind
Consortium, and performed a schizophrenia rTWAS using GWAS summary statistics from an African
American cohort. Unfortunately, GWAS from non-European cohorts remain severely underpowered, which
justifies the fact that we found no Bonferroni significant expression signatures associated with schizophrenia
in that analysis. Nevertheless, we are very excited to have created the African rTWAS weights and to make
them freely available to the scientific community for future work.

Regarding point (2), about the use of cases in the construction of SNP weights, we were not concerned
about this because Gusev et al. (2018) found consistent *cis*-regulatory mechanisms governing gene
expression across cases and controls in the dataset analysed in your study. However, in light of your
comment, we added a new Results section (*“Sensitivity analyses”*, line 370). We found evidence that adding
cases and thus improving sample size by 133% increased power to detect genes with a *cis*-heritable
expression component by 85% as well as rTWAS association signals by 19%, while maintaining the
specificity of the rTWAS results (Pearson’s $r = 0.95$, $P < 2.2 \times 10^{-16}$).

#

**# 3. In the Discussion section, the author reported that the HERV families they identified are**
**# different from those previously reported in the literature. Can the authors elaborate more on**
**# this inconsistency? What might contribute to it?**

#

Thank you for this comment. We have rephrased our manuscript extensively and made sure to discuss the
factors that we believe have contributed to perpetuate the inconsistencies observed in HERV research in
general (in the *Introduction*), as well as the reasons that likely explain differences observed between our
study and previous studies.

In the Introduction, we first highlight how previous research has mostly employed methods to assess HERV
expression which lack genomic specificity, and that rely on analysis of small case-control studies:

- • *“... most studies precede the comprehensive genomic annotation of these sequences. They also*
*relied on methods that aggregate family-level expression data, such as Western blotting, reverse*
*transcriptase quantitative PCR (RT-qPCR), or microarrays. Most also analysed very small sample*
*sizes, meaning they were underpowered for the investigation of complex polygenic traits (Bray &*
*O’Donovan, 2018). Finally, by employing case-control study designs, they were more likely to*
*capture expression changes elicited by environmental factors associated with a psychiatric*
*diagnosis, such as smoking or treatment (Gusev et al., 2016)”*

In the Discussion, we then cite the potential reasons that may explain why our study identified HERVs with
relevance to psychiatry from families that are different from those previously highlighted in association with
the psychiatric conditions tested:

- • *“The probable reason for such discrepancy is that earlier studies employed HERV quantification*
*methods that averaged expression signals from across multiple HERV copies in the genome (as*
*discussed in the Introduction). In addition, because these studies aimed to detect case-control*
*differences in samples originating from small cohorts, they were more likely to detect secondary*
*disease expression signatures, including those associated with effects of medication or smoking. Our*
*work, on the other hand, used a specialised HERV expression quantification approach that infers*
*HERV expression levels with genomic precision. It also focuses on expression signatures associated*
*with genetic risk and thus mechanisms more likely to be implicated in disorder aetiology.”*

#
#
**# 4. Related to the comment above, part of the inconsistency could be a result of the still-small**
**# sample size of this study. I suggest that the authors perform a statistical power analysis of**
**# the rTWAS.**
#

Thank you for this comment. To provide some clarification, statistical power for a TWAS is mostly a factor of
the size of the GWAS analysed, although power will also be impacted by a genetic feature’s expression
heritability. Detection of genes (or HERVs) with a cis-heritable component will depend on the size of the
cohort used to calculate the SNP weights, which is something we demonstrated in the new version of the
manuscript, when comparing the number of genetic features with a cis-heritable expression component in
the full European cohort (N = 563) vs. in the European cohort of unaffected individuals only (N = 242). For
instance, we found reductions of 46% and 27% in the number of HERVs and protein coding genes with a cis-
heritable expression component in the reduced sample, relative to the entire cohort. Work from He et al.
(2022) further suggests that the detection of genetic features with a cis-heritable expression component
reaches a plateau after the addition of a few hundred samples.

As mentioned in our response above, the inconsistencies between our findings and previous reports in the
literature are more likely to be related to the completely different methodologies and study designs
employed, rather than being related to an issue of power. For instance, early HERV studies employed case-
control designs using small cohorts where HERV expression was assessed using methods that lacked
genomic precision (microarrays, Western blotting, RT-qPCR). Our study, on the other hand, is assessing the
effect of genotype (not case-control status) at multiple genetic variants associated with psychiatric disorders
on the expression of 4,289 HERVs expressed in the brain, which have been mapped to their source genomic
locations and individually quantified.

- Akrami, R., Jacobsen, A., Hoell, J., Schultz, N., Sander, C., & Larsson, E. (2013). Comprehensive analysis of
long non-coding RNAs in ovarian cancer reveals global patterns and targeted DNA amplification.
*PLOS ONE*, 8(11), e80306. <https://doi.org/10.1371/journal.pone.0080306>
- Bray, N. J., & O'Donovan, M. C. (2018). The genetics of neuropsychiatric disorders. *Brain and Neuroscience*
*Advances*, 2, 2398212818799271. <https://doi.org/10.1177/2398212818799271>
- Du, Z., Fei, T., Verhaak, R. G., Su, Z., Zhang, Y., Brown, M., Chen, Y., & Liu, X. S. (2013). Integrative
genomic analyses reveal clinically relevant long noncoding RNAs in human cancer. *Nat Struct Mol*
*Biol*, 20(7), 908-913. <https://doi.org/10.1038/nsmb.2591>
- Esteller, M. (2011). Non-coding RNAs in human disease. *Nat Rev Genet*, 12(12), 861-874.
<https://doi.org/10.1038/nrg3074>
- Fatica, A., & Bozzoni, I. (2014). Long non-coding RNAs: new players in cell differentiation and development.
*Nat Rev Genet*, 15(1), 7-21. <https://doi.org/10.1038/nrg3606>
- GTEx Consortium, Aguet, F., Brown, A. A., Castel, S. E., Davis, J. R., He, Y., Jo, B., Mohammadi, P., Park, Y.,
Parsana, P., Segrè, A. V., Strober, B. J., Zappala, Z., Cummings, B. B., Gelfand, E. T., Hadley, K.,
Huang, K. H., Lek, M., Li, X., . . . Zhu, J. (2017). Genetic effects on gene expression across human
tissues [Article]. *Nature*, 550, 204. <https://doi.org/10.1038/nature24277>
- Guffanti, G., Bartlett, A., Klengel, T., Klengel, C., Hunter, R., Glinsky, G., & Macciardi, F. (2018). Novel
Bioinformatics Approach Identifies Transcriptional Profiles of Lineage-Specific Transposable
Elements at Distinct Loci in the Human Dorsolateral Prefrontal Cortex. *Molecular Biology and*
*Evolution*, 35(10), 2435-2453. <https://doi.org/10.1093/molbev/msy143>
- Gusev, A., Ko, A., Shi, H., Bhatia, G., Chung, W., Penninx, B. W., Jansen, R., de Geus, E. J., Boomsma, D.
I., Wright, F. A., Sullivan, P. F., Nikkola, E., Alvarez, M., Civelek, M., Lusi, A. J., Lehtimäki, T.,
Raitoharju, E., Kahonen, M., Seppala, I., . . . Pasaniuc, B. (2016). Integrative approaches for large-
scale transcriptome-wide association studies. *Nat Genet*, 48(3), 245-252.
<https://doi.org/10.1038/ng.3506>
- Gusev, A., Mancuso, N., Won, H., Kousi, M., Finucane, H. K., Reshef, Y., Song, L., Safi, A., McCarroll, S.,
Neale, B. M., Ophoff, R. A., O'Donovan, M. C., Crawford, G. E., Geschwind, D. H., Katsanis, N.,
Sullivan, P. F., Pasaniuc, B., Price, A. L., & Schizophrenia Working Group of the Psychiatric
Genomics, C. (2018). Transcriptome-wide association study of schizophrenia and chromatin activity
yields mechanistic disease insights. *Nature Genetics*, 50(4), 538-548.
<https://doi.org/10.1038/s41588-018-0092-1>
- He, R., Xue, H., & Pan, W. (2022). Statistical power of transcriptome-wide association studies. *Genet*
*Epidemiol*, 46(8), 572-588. <https://doi.org/10.1002/gepi.22491>
- Iñiguez, L. P., de Mulder Rougvié, M., Stearrett, N., Jones, R. B., Ormsby, C. E., Reyes-Terán, G., Crandall,
326 K. A., Nixon, D. F., & Bendall, M. L. (2019). Transcriptomic analysis of human endogenous
retroviruses in systemic lupus erythematosus. *Proc Natl Acad Sci U S A*, 116(43), 21350-21351.
<https://doi.org/10.1073/pnas.1907705116>
- Langfelder, P., & Horvath, S. (2008). WGCNA: an R package for weighted correlation network analysis. *BMC*
*Bioinformatics*, 9(1), 559. <https://doi.org/10.1186/1471-2105-9-559>
- Levey, D. F., Stein, M. B., Wendt, F. R., Pathak, G. A., Zhou, H., Aslan, M., Quaden, R., Harrington, K. M.,
Nuñez, Y. Z., Overstreet, C., Radhakrishnan, K., Sanacora, G., McIntosh, A. M., Shi, J.,
Shringarpure, S. S., Concato, J., Polimanti, R., & Gelernter, J. (2021). Bi-ancestral depression
GWAS in the Million Veteran Program and meta-analysis in >1.2 million individuals highlight new
therapeutic directions. *Nat Neurosci*, 24(7), 954-963. <https://doi.org/10.1038/s41593-021-00860-2>
- Neulinger-Munoz, M., Schaack, D., Grekova, S. P., Bauer, A. S., Giese, T., Salg, G. A., Espinet, E., Leuchs,
B., Heller, A., Nuesch, J. P. F., Schenk, M., Volkmar, M., & Giese, N. A. (2021). Human
Retrotransposons and the Global Shutdown of Homeostatic Innate Immunity by Oncolytic Parvovirus
H-1PV in Pancreatic Cancer. *Viruses*, 13(6). <https://doi.org/10.3390/v13061019>
- Sekar, A., Bialas, A. R., de Rivera, H., Davis, A., Hammond, T. R., Kamitaki, N., Tooley, K., Presumey, J.,
Baum, M., Van Doren, V., Genovese, G., Rose, S. A., Handsaker, R. E., Daly, M. J., Carroll, M. C.,
Stevens, B., & McCarroll, S. A. (2016). Schizophrenia risk from complex variation of complement
component 4. *Nature*, 530(7589), 177-183. <https://doi.org/10.1038/nature16549>
- Stearrett, N., Dawson, T., Rahnavard, A., Bachali, P., Bendall, M. L., Zeng, C., Caricchio, R., Pérez-Losada,
345 M., Grammer, A. C., Lipsky, P. E., & Crandall, K. A. (2021). Expression of Human Endogenous
Retroviruses in Systemic Lupus Erythematosus: Multiomic Integration With Gene Expression
[Original Research]. *Frontiers in immunology*, 12. <https://doi.org/10.3389/fimmu.2021.661437>

Reviewer #1 (Remarks to the Author):

Dear Authors,

Thank you for considering my comments and revising your manuscript accordingly. I recommend the manuscript for publication.

Specific feedback:

#Use of total RNA vs poly A+ library:

I understand your rationale for using total RNA to analyze HERV expression. For even stronger analysis, I recommend including the number of expressed HERV elements in (1) intron of expressed genes introns, (2) intron of unexpressed genes, and (3) extragenic regions. This will provide valuable evidence for pervasive transcription.

#Coregulation genes/TEs:

Thank you for sharing your long-read sequencing data, it is very promising.

#Chimeric transcripts etc.:

I agree that exploring chimeric transcripts might be beyond the scope of this current manuscript. However, I appreciate you offering the information that such analysis is possible with short-read sequencing, especially for HERV elements. This could be valuable for future research.

#4 Figures

Thank you for improving the figures and the legends.

Reviewer #2 (Remarks to the Author):

The authors have adequately addressed my concerns.

Reviewer #2 (Remarks on code availability):

Codes are organized, with sufficient details provided.

##### Reviewer #1 (Remarks to the Author):
# Dear Authors,
# Thank you for considering my comments and revising your manuscript accordingly.
# I recommend the manuscript for publication.
# Specific feedback:
## - Use of total RNA vs poly A+ library:
# I understand your rationale for using total RNA to analyze HERV expression.
# For even stronger analysis, I recommend including the number of expressed HERV
# elements in (1) intron of expressed genes introns, (2) intron of unexpressed
# genes, and (3) extragenic regions. This will provide valuable evidence for
# pervasive transcription.
## - Coregulation genes/TEs:
# Thank you for sharing your long-read sequencing data, it is very promising.
## - Chimeric transcripts etc.:
# I agree that exploring chimeric transcripts might be beyond the scope of this
# current manuscript. However, I appreciate you offering the information that such
# analysis is possible with short-read sequencing, especially for HERV elements.
# This could be valuable for future research.
## - Figures
# Thank you for improving the figures and the legends.
#

Dear Reviewer 1, thank you once again for reviewing our manuscript and for your kind comments
on our work. In the revised version of the manuscript, we have included a new Figure (Figure 5B),
which addresses your comment about *“the number of expressed HERV elements in (1) intron of*
*expressed genes introns, (2) intron of unexpressed genes, and (3) extragenic regions”*.

Regarding your second comment, about *“offering the information that such analysis is possible*
*[about detection of chimeric HERV transcripts using short-read sequencing]”*, we have included this
sentence in the Discussion (lines 400-403), as per your suggestion:

- “While it is possible to identify chimeric HERV transcripts using short-read RNA-
sequencing, long-read RNA-sequencing studies are likely to be better equipped to identify
transcripts originating from repetitive sequences.”

#
##### Reviewer #2 (Remarks to the Author):
# The authors have adequately addressed my concerns.
# Reviewer #2 (Remarks on code availability):
# Codes are organized, with sufficient details provided.
#

Dear Reviewer 2, thank you for your kind assessment of our work.